# Modeling the impact of racial and ethnic disparities on COVID-19 epidemic dynamics

Kevin C Ma[1]*, Tigist F Menkir[2], Stephen Kissler[1], Yonatan H Grad[1,3], Marc Lipsitch[1,2]

[1]Department of Immunology and Infectious Diseases, Harvard TH Chan School of Public Health, Boston, United States; [2]Center for Communicable Disease Dynamics, Department of Epidemiology, Harvard TH Chan School of Public Health, Boston, United States; [3]Division of Infectious Diseases, Brigham and Women's Hospital and Harvard Medical School, Boston, United States

## Abstract

**Background:** The impact of variable infection risk by race and ethnicity on the dynamics of SARS-CoV-2 spread is largely unknown.

**Methods:** Here, we fit structured compartmental models to seroprevalence data from New York State and analyze how herd immunity thresholds (HITs), final sizes, and epidemic risk change across groups.

**Results:** A simple model where interactions occur proportionally to contact rates reduced the HIT, but more realistic models of preferential mixing within groups increased the threshold toward the value observed in homogeneous populations. Across all models, the burden of infection fell disproportionately on minority populations: in a model fit to Long Island serosurvey and census data, 81% of Hispanics or Latinos were infected when the HIT was reached compared to 34% of non-Hispanic whites.

**Conclusions:** Our findings, which are meant to be illustrative and not best estimates, demonstrate how racial and ethnic disparities can impact epidemic trajectories and result in unequal distributions of SARS-CoV-2 infection.

**Funding:** K.C.M. was supported by National Science Foundation GRFP grant DGE1745303. Y.H.G. and M.L. were funded by the Morris-Singer Foundation. M.L. was supported by SeroNet cooperative agreement U01 CA261277.

*For correspondence:
kevinchenma@g.harvard.edu

## Introduction

The dynamics of SARS-CoV-2 spread are influenced by population heterogeneity. This is especially true for herd immunity, which occurs when susceptible individuals in a population are indirectly protected from infection due to immunity in others. The herd immunity threshold (HIT) is the fraction of the population that is non-susceptible when an unmitigated epidemic reaches its peak, and estimating the HIT for SARS-CoV-2 is important for forecasting the harm associated with letting the epidemic spread in the absence of interventions (*Randolph and Barreiro, 2020*). A population that has reached the HIT is protected from a new epidemic occurring, until births or waning immunity reduce the proportion nonsusceptible below the HIT, but existing cases will still lead to some onward transmission as the epidemic declines and can often result in a final epidemic size that exceeds the HIT. In a population with homogeneous mixing, the HIT is $1-1/R_0$, where $R_0$ is the basic reproduction number; this translates to an HIT of 67% using an $R_0$ of 3.

However, population homogeneity is an unrealistic assumption, and models incorporating heterogeneity in social exposure and infection susceptibility (defined as the probability of infection given exposure) generally result in lowered HITs (*Hill and Longini, 2003*; *Britton et al., 2020*; *Gomes et al., 2020*; *Aguas et al., 2020*; *Tkachenko et al., 2020*). The key idea behind these models is that subpopulations important for epidemic spread (i.e., those with substantially increased susceptibility or exposure) become infected – and thus develop immunity – early on in an epidemic's course. Herd immunity for the population overall is then achieved earlier because once these individuals are no longer susceptible to infection, further epidemic spread is slowed.

Importantly, these models also imply that in locations where SARS-CoV-2 has spread there may be demographic subpopulations with particularly high cumulative incidences of infection due to increased exposure, susceptibility, or both. Seroprevalence studies – which characterize past exposure by identifying SARS-CoV-2 antibodies – can identify these subpopulations and are more reliable and unbiased than case data, which suffer from under-reporting and other biases (*Metcalf et al., 2020*). Identifying and building structured models with these groups in mind is important for understanding how variation in exposure or susceptibility and social disparities is interconnected. These models are also useful for designing interventions that can both reduce disparities and disrupt overall transmission by focusing efforts on groups most affected by high transmission rates (*Wallinga et al., 2010*; *Bubar et al., 2021*). Transmission models in this space have incorporated subpopulation structure by focusing primarily on accounting for variation in susceptibility and exposure by age (*Britton et al., 2020*; *Davies et al., 2020*; *Miller et al., 2020*). Supporting this approach, susceptibility to infection, contact rates, and cumulative incidence in some locations all appear to vary by age (*Davies et al., 2020*; *Mossong et al., 2008*). Nonetheless, serosurveys in Belgium, Spain, Iran, New York City (NYC), Brazil, and other places exhibit relatively low variation in seropositivity by age (*Herzog et al., 2020*; *Pollán et al., 2020*; *Shakiba et al., 2020*; *Rosenberg et al., 2020*; *Hallal et al., 2020*), indicating additional factors that govern transmission spread.

Substantial racial and ethnic disparities in infection rates, hospitalizations, and deaths have been characterized across the US (*Chamie et al., 2020*; *Moore et al., 2020*; *Millett et al., 2020a*; *Pan et al., 2020*; *Chen and Krieger, 2020*; *Bassett et al., 2020*; *Hanage et al., 2020*), but it is unclear how these heterogeneities in risk are expected to change over time and what implications – if any – they have on overall epidemic dynamics. Here, we aim to address these questions by fitting compartmental SEIR transmission models structured by race and ethnicity to seroprevalence data from NYC and Long Island (*Rosenberg et al., 2020*). We focus primarily on building and analyzing variable exposure models because observed disparities in infection rates in US cities are strongly attributable to differences in mobility and exposure (*Chang et al., 2021*; *Zelner et al., 2020*; *Kissler et al., 2020*). Because of the challenges in acquiring racial and ethnic COVID-19 data (*Krieger et al., 2020b*), including social contact data that can be used in transmission models, we analyze a range of model structures that are compatible with the data and assess how these assumptions affect estimates of HITs, final epidemic sizes, and longitudinal trends in risk across groups. These results highlight the importance of developing socially informed COVID-19 transmission models that incorporate patterns of epidemic spread across racial and ethnic groups.

## Materials and methods
### SEIR model
We initially modeled transmission dynamics in a homogeneous population using a SEIR compartmental SARS-CoV-2 infection model:

$$\frac{dS}{dt} = -\beta IS \tag{1}$$

$$\frac{dE}{dt} = \beta IS - rE \tag{2}$$

$$\frac{dI}{dt} = rE - \gamma I \tag{3}$$

$$\frac{dR}{dt} = \gamma I \tag{4}$$

where $S, E, I, R$ refer to the number of people in susceptible, latently infected, infectious, and recovered compartments, respectively. Given a mean incubation period and mean serial interval of 5 days as suggested by empirical studies (*Nishiura et al., 2020*; *Lauer et al., 2020*), we set the mean latent

period $1/r$ to be 3 days to allow for pre-symptomatic transmission and the mean infectious period $1/\gamma$ to be 4 days to coincide with the observed serial interval. The per capita transmission rate is given by $\beta = R_0\gamma/N$, where $N$ is the total number of people in the population.

We extended this model to incorporate multiple racial and ethnic groups by including SEIR compartmental variables for each group, which interact through a social contact matrix that governs the interactions between and within groups. In matrix form, the structured SEIR model is given by

$$\frac{d\mathbf{S}}{dt} = -(\mathbf{BI}) \circ \mathbf{S} \tag{5}$$

$$\frac{d\mathbf{E}}{dt} = (\mathbf{BI}) \circ \mathbf{S} - r\mathbf{E} \tag{6}$$

$$\frac{d\mathbf{I}}{dt} = r\mathbf{E} - \gamma\mathbf{I} \tag{7}$$

$$\frac{d\mathbf{R}}{dt} = \gamma\mathbf{I} \tag{8}$$

where $\circ$ denotes element-wise multiplication and $\mathbf{S}, \mathbf{E}, \mathbf{I}, \mathbf{R}$ are column vectors comprising the compartmental variables for each group (e.g., $\mathbf{S} = [S_0, ..., S_p]^T$ for $p$ demographic groups). We let $S_0$ denote non-Hispanic whites, $S_1$ denote Hispanics or Latinos, $S_2$ denote non-Hispanic African-Americans, $S_3$ denote non-Hispanic Asians, and $S_4$ denote multiracial or other demographic groups, with similar ordering for elements in vectors $\mathbf{E}$ through $\mathbf{R}$.

We define contacts to be interactions between individuals that allow for transmission of SARS-CoV-2 with some non-zero probability. Following the convention for age-structured transmission models (*Wallinga et al., 2019*), we defined the $p \times p$ per capita social contact matrix $\mathbf{C}$ to consist of elements $c_{i \leftarrow j}$ at row and column $j$, representing the per capita rate that individuals from group $i$ are contacted by individuals of group $j$. Letting $N_i$ be the total number of individuals in group $i$, the social contact matrix $\mathbf{M}$ consists of elements $m_{i \leftarrow j} = c_{i \leftarrow j} * N_i$, which represents the average number of individuals in group $i$ encountered by an individual in group $j$. The susceptibility to infection can vary between groups, which we modeled by allowing the probability of infection given contact with an infected individual to vary: $\mathbf{q} = [q_0, ..., q_p]^T$. The transmission matrix $\mathbf{B}$ is then given by $(\mathbf{q1}^T) \circ \mathbf{C}$, where $1^T$ is a one by $p$ vector of 1 s:

$$\mathbf{B} = \begin{bmatrix} q_0 & ... & q_0 & ... & q_0 \\ \vdots & & \vdots & & \vdots \\ q_4 & ... & q_4 & ... & q_4 \end{bmatrix} \circ \begin{bmatrix} c_{0 \leftarrow 0} & ... & c_{0 \leftarrow 2} & ... & c_{0 \leftarrow 4} \\ \vdots & & \vdots & & \vdots \\ c_{4 \leftarrow 0} & ... & c_{4 \leftarrow 2} & ... & c_{4 \leftarrow 4} \end{bmatrix} \tag{9}$$

Given mean duration of infectiousness $1/\gamma$, the next-generation matrix $\mathbf{G}$, representing the average number of infections in group $i$ caused by an infected individual in group $j$, is given by $(q1^T) \circ \mathbf{M}/\gamma = \mathbf{N} \circ \mathbf{B}/\gamma$. $R_0$ for the overall population described by this structured model was calculated by computing the dominant eigenvalue of matrix $\mathbf{G}$. The effective reproduction number $R_t$ at time $t$ was calculated by computing the dominant eigenvalue of $\mathbf{G_t} = (\mathbf{q}1^T) \circ \mathbf{M_t}/\gamma = \mathbf{S_t} \circ \mathbf{B}/\gamma$, where the elements in $\mathbf{M_t}$ are given by $c_{i \leftarrow j} * S_{i,t}$ and $S_{i,t}$ is the number of susceptible individuals in group $i$ at time $t$. To hold $R_0$ values across model types constant when calculating HITs and final epidemic sizes, we re-scaled transmission matrices to have the same dominant eigenvalue. We also calculated the instantaneous incidence rate at some time $t$ for all groups by calculating the force of infection $\lambda_t = (\mathbf{BI_t}) \circ \mathbf{S_t}$. To account for the effects of social distancing and other non-pharmaceutical interventions (NPIs), we scaled the transmission rate by a factor $\alpha$ beginning when 5% cumulative incidence in the population was reached, representing an established and expanding epidemic, for a variable duration. We analyzed a range of $\alpha$ values to reflect the variation in NPIs implemented.

## Structured model variants

Simplifying assumptions are needed to constrain the number of variables to estimate in $\mathbf{B}$ given limited data. Under the *variable susceptibility* model, we set the contact rates $c_{i \leftarrow j}$ to all be 1, indicating no heterogeneity in exposure, but allowed the $q_j$ in the susceptibility vector to vary (i.e., $\mathbf{B} = \mathbf{q}1^T$).

Under each of two variable exposure models, in contrast, we set the susceptibility factors $q_j$ to be equal. The simplest variable exposure model we analyzed was the *proportionate mixing* model, which assumes that the contact rate for each pair of groups is proportional to the total contact rate of the two groups (i.e., total number of contacts per unit time for an individual of group $i$) (*Hethcote, 1996*).

Denoting $a_i$ as the total contact rate for a member of group and **a** as the $1 \times p$ vector of $a_i$s, the $ij$ th entry in the transmission matrix is given by

$$\beta_{i \leftarrow j} = q \frac{a_i a_j}{\sum_k a_k N_k} \tag{10}$$

and the overall transmission matrix **B** can be written as

$$\mathbf{B} = \frac{q}{\sum_k a_k N_k} \mathbf{a} \mathbf{a}^T \tag{11}$$

Finally, under the *assortative mixing* assumption, we extended this model by partitioning a fraction $\epsilon$ of contacts to be exclusively within-group and distributed the rest of the contacts according to proportionate mixing (with $\delta_{ij}$ being an indicator variable that is 1 when $i = j$ and 0 otherwise) (*Hethcote, 1996*):

$$\beta_{i \leftarrow j} = (1 - \epsilon) q \frac{a_i a_j}{\sum_k a_k N_k} + \epsilon \delta_{ij} q \frac{a_i}{N_i} \tag{12}$$

$$\mathbf{B} = \frac{(1-\epsilon)q}{\sum_k a_k N_k} \mathbf{a} \mathbf{a}^T + \epsilon\, q \, diag(\mathbf{a} \circ 1/\mathbf{N}) \tag{13}$$

## Calculating the HIT and final epidemic size

We defined the HIT for all models as the fraction of nonsusceptible people when the effective reproduction number $R_t$ first crosses 1. In the homogenous model, where $R_t = S_t \beta / \gamma$, the analytical solution for the HIT occurs when the fraction of nonsusceptible individuals equals $1-1/R_0$. In the structured models of heterogeneous populations, the HIT was calculated via simulation: we took the dominant eigenvalue of $\mathbf{G_t}$ at each timestep to calculate $R_t$ and identified the number of nonsusceptible individuals when $R_t$ first decreased below 1. For the heterogeneous models with mitigation measures, $R_t$ was calculated at each timestep with respect to the corresponding unmitigated epidemic; in other words, the mitigation scaling factor $\alpha$ was not included in the $R_t$ calculation. This identifies the point in the epidemic trajectory at which the population reaches the HIT even if all mitigation measures were lifted (i.e., HIT due to population immunity), as opposed to the point in the trajectory when the population transiently reaches the HIT due to mitigation measures (see *Figure 1—figure supplement 1* for further explanation).

Final epidemic sizes were calculated by simulation by running the epidemic out to 1 year for $R_0$ above 2 and 4 years for $R_0$ below 2 to allow additional time for the slower epidemics to fully resolve. The final time point was used as the estimate for the final epidemic size.

## Model fitting and data sources

SEIR differential equations were solved using the lsoda function in the deSolve package (version 1.28) of R (version 3.6.3). We estimated the $a_i$ in the variable exposure models and the $q_i$ in the variable susceptibility models using maximum likelihood fits to a single cross-sectional serosurvey from New York, which was collected from over 15,000 adults in grocery stores from April 19 to 28 (*Rosenberg et al., 2020*). We assumed that the seroprevalence data (adjusted cumulative incidence estimates from Table 2 in *Rosenberg et al., 2020*) were collected via a binomial sampling process: at a given time point $t_s$ representing the time of the serosurvey, the number of seropositive cases $Y_i(t_s)$ in group $i$ is distributed $Bin(m_i, R_i(t_s)/N_i(t_s))$, where $m_i$ is the number of people tested from group $i$ in the serosurvey and $R_i/N_i$ is the fraction of recovered people from the SEIR model. The likelihood was calculated jointly for all demographic groups, with $t_s$ set to 100 days and the initial number of infected individuals set to 1 in each demographic group.

We conducted sensitivity analyses to assess whether these assumptions on epidemic timing, and number and distribution of initial infected individuals, affected parameter and HIT estimates. Varying the timing of epidemic start did not substantially affect HIT estimates as long as the time between epidemic start and serosurvey $t_s$ was reasonably large (e.g., >20 days) and assortativity was low ($\epsilon < 0.8$) (*Figure 2—figure supplement 1*). The distribution and number of initial infected individuals also did not substantially affect HIT estimates for low levels of assortativity ($\epsilon < 0.8$) (*Figure 2—figure supplements 2 and 3*). We limited our analyses to models with $\epsilon$ less than 0.8.

We acquired total population numbers (i.e., $N_i$ for $i \in \{0, ..., 4\}$) from the 2018 ACS census 1-year estimates Table B03002 (Hispanic or Latino origin by race) subsetted to the following counties: Bronx,

Kings, New York, Queens, and Richmond Counties for NYC and Nassau and Suffolk Counties for Long Island. We acquired population numbers at the level of 'all block groups' within the above counties from the 2018 ACS 5-year estimates Table B03002 (Hispanic or Latino origin by race). Copies of the census data we used are available at https://github.com/kevincma/covid19-race-ethnicity-model/tree/main/data.

## Census-informed transmission model

The total number of contacts $C_{ij}$ between groups $i$ and $j$ can be calculated using the assortative mixing social contact matrix.

$$C_{ij} = c_{ij} N_i N_j = (1 - \epsilon) \frac{a_i a_j N_i N_j}{\sum_k a_k N_k} + \epsilon \delta_{ij} a_i N_j$$

To fit this assortative mixing model to both serosurvey and census data, we modeled interactions between racial and ethnic groups at the census block group level – which we interpreted to be roughly equivalent to neighborhoods – allowing the structure of the census data to inform the dynamics of transmission. Specifically, we assumed proportionate mixing between racial and ethnic groups in each census block group, with no interactions between block groups. Under this *proportionate mixing within neighborhoods* assumption, the total number of contacts $C'_{ij}$ between groups $i$ and $j$ is proportional to:

$$C'_{ij} \propto \sum_l^L a_i a_j N_{i,l} N_{j,l}$$

where $L$ is the number of census block groups, $N_{j,l}$ is the number of people from demographic group $j$ in census block $l$, and $a_j$ is the total contact rate per individual in group $j$ as before. Within each neighborhood, proportionate mixing holds: the total number of contacts between two groups is proportional to the activity level and neighborhood population of the groups. Additionally, similarly to the previous models, the total number of contacts across all groups ($\sum_{ij} C'_{ij}$) must equal $\sum_k a_k N_k$; to satisfy this constraint, we set a proportionality constant:

$$C'_{ij} = \frac{\sum_k a_k N_k}{\sum_{ijl} a_i a_j N_{i,l} N_{j,l}} \sum_l^L a_i a_j N_{i,l} N_{j,l}$$

To fit $\epsilon$, we minimized the absolute difference between these two formulations ($C_{ij}$ and $C'_{ij}$) of the total number of contacts across all pairs of groups (*Figure 2—figure supplement 4*):

$$\hat{\epsilon} = \underset{\epsilon}{\mathrm{argmin}} \sum_{ij} |C_{ij} - C'_{ij}| =$$
$$\mathrm{argmin}_\epsilon \sum_{ij} |(1 - \epsilon) \frac{a_i a_j N_i N_j}{\sum_k a_k N_k} + \epsilon \delta_{ij} a_i N_j - \frac{\sum_k a_k N_k}{\sum_{ijl} a_i a_j N_{i,l} N_{j,l}} \sum_l^L a_i a_j N_{i,l} N_{j,l}|$$

Using the fitted value $\hat{\epsilon}$, we then conducted maximum likelihood to fit the varying $a_i$ as described previously. We repeated this process iteratively – holding $a_i$ constant while $\epsilon$ was fit, and then vice versa – until convergence, which we defined as the difference between successive $\hat{\epsilon}$ values being lower than a threshold of 0.001 (*Supplementary file 3*). For the first iteration, we fit $\hat{\epsilon}$ holding $a_i$ constant at 1. We used this iterative fitting procedure to accommodate both the seroprevalence and census data because the single seroprevalence time point cannot fit both the activity levels and $\epsilon$.

To empirically characterize average neighborhood composition from the census data, we also calculated the exposure index matrix $P$ with elements $P_{ij}$ for demographic groups $i$ and $j$, defined similarly to *McCauley et al., 2001*; *Richardson et al., 2020*

$$P_{ij} = \sum_l^L \left( \frac{N_{j,l}}{N_j} \right) \left( \frac{N_{i,l}}{T_l} \right) \tag{14}$$

where $T_l$ is the total number of people in census block group $l$ and other variables are defined as before. The exposure indices were used for descriptive purposes only and not used in the model fitting approach.

## Variable susceptibility versus variable exposure models

The serosurvey data were compatible with variable susceptibility models in which Hispanics or Latinos, non-Hispanic Black people, non-Hispanic Asians, and multiracial or other people had 2.25, 1.62, 0.86,

and 1.28 times the susceptibility to infection relative to non-Hispanic whites in NYC, respectively, and 4.32, 1.96, 0.92, and 2.48 times the susceptibility to infection relative to non-Hispanic whites in Long Island, respectively. As with variable exposure models, these differences in susceptibility lowered herd immunity levels and final epidemic sizes relative to the homogeneous model (*Figure 1—figure supplement 2*), but to a lesser extent; for instance, variable susceptibility models resulted in HITs ~10% greater than HITs under proportionate mixing for Long Island.

The difference between these models is that incorporating heterogeneity in susceptibility only affects susceptible individuals, but heterogeneity in exposure impacts both susceptible and infectious individuals: individuals from racial and ethnic groups with higher contact rates are both more likely to be infected, and when infected, to infect a greater number of secondary cases. This contrast is clear when comparing the next-generation matrices for each model, which lists the average number of secondary infections caused by an infected individual from a given demographic group (*Figure 2—figure supplements 5 and 6*). The epidemic resolves at an earlier stage in variable exposure models once these key transmission groups become immune because of this additional compound effect on transmission.

Our results contrasting mechanistic variable exposure and susceptibility models are in line with theoretical studies, which also indicate that models incorporating heterogeneity in exposure have more pronounced effects on HITs than models incorporating heterogeneity in susceptibility, assuming comparable continuous distributions of exposure and susceptibility (*Tkachenko et al., 2020*; *Gomes et al., 2020*). Tkachenko et al. showed that $HIT = 1 - (1/R_0)^{(1/\lambda)}$, where $\lambda$ is either $1 + CV^2$ for variable susceptibility models or $1 + CV^2(2 + \gamma_s CV)/(1 + CV^2)$ for variable exposure models, and $CV$ is the coefficient of variation and $\gamma_s$ is the skewness for the exposure distribution (*Tkachenko et al., 2020*). We calculated CV and skewness using the susceptibility and total contact rate ratios and substituted those values into the HIT formula, which is an approximation because our exposure and susceptibility distributions are discrete; nonetheless, the approximations result in similar HIT curves to the simulation results (*Figure 1—figure supplement 3*).

## Code availability

Code and data to reproduce all analyses and figures are available at https://github.com/kevincma/covid19-race-ethnicity-model *Ma et al., 2021*, copy archived at swh:1:rev:75574621317a599e90 58236f62bb34de63120e99. An executable version of the Jupyter notebook is available at https:// mybinder.org/v2/gh/kevincma/covid19-race-ethnicity-model/HEAD.

## Results

We model the dynamics of COVID-19 infection allowing for social exposure to infection to vary across racial and ethnic groups. Models incorporating variable susceptibility to COVID-19 are commonly used when stratifying by age because children are thought to have decreased susceptibility to infection (*Davies et al., 2020*). Variable susceptibility to infection across racial and ethnic groups has been less well characterized, and observed disparities in infection rates can already be largely explained by differences in mobility and exposure (*Chang et al., 2021*; *Zelner et al., 2020*; *Kissler et al., 2020*), likely attributable to social factors such as structural racism that have put racial and ethnic minorities in disadvantaged positions (e.g., employment as frontline workers and residence in overcrowded, multi-generational homes) (*Henry Akintobi et al., 2020*; *Thakur et al., 2020*; *Tai et al., 2021*; *Khazanchi et al., 2020*). In line with the notion that variation in exposure could instead be the main driver of observed seroprevalence differences, our primary focus is on analyzing variable exposure models; we have also analyzed variable susceptibility models for comparison (see Materials and methods section).

The simplest variable exposure models assume proportionate mixing, where the contact rate between groups is set to be proportional to the total contact rates (i.e., total number of contacts per time period per individual) of the two groups (*Hethcote, 1996*). We fit proportionate mixing models allowing for variable contact rates across racial and ethnic demographic groups to serosurvey data collected in late April from NYC and Long Island, comprising 5946 and 2074 adults, respectively (*Rosenberg et al., 2020*). The serosurvey data were compatible with proportionate mixing models in which Hispanics or Latinos, non-Hispanic Black people, non-Hispanic Asians, and multiracial or other people had 2.25, 1.62, 0.86, and 1.28 times the total contact rates relative to non-Hispanic whites in

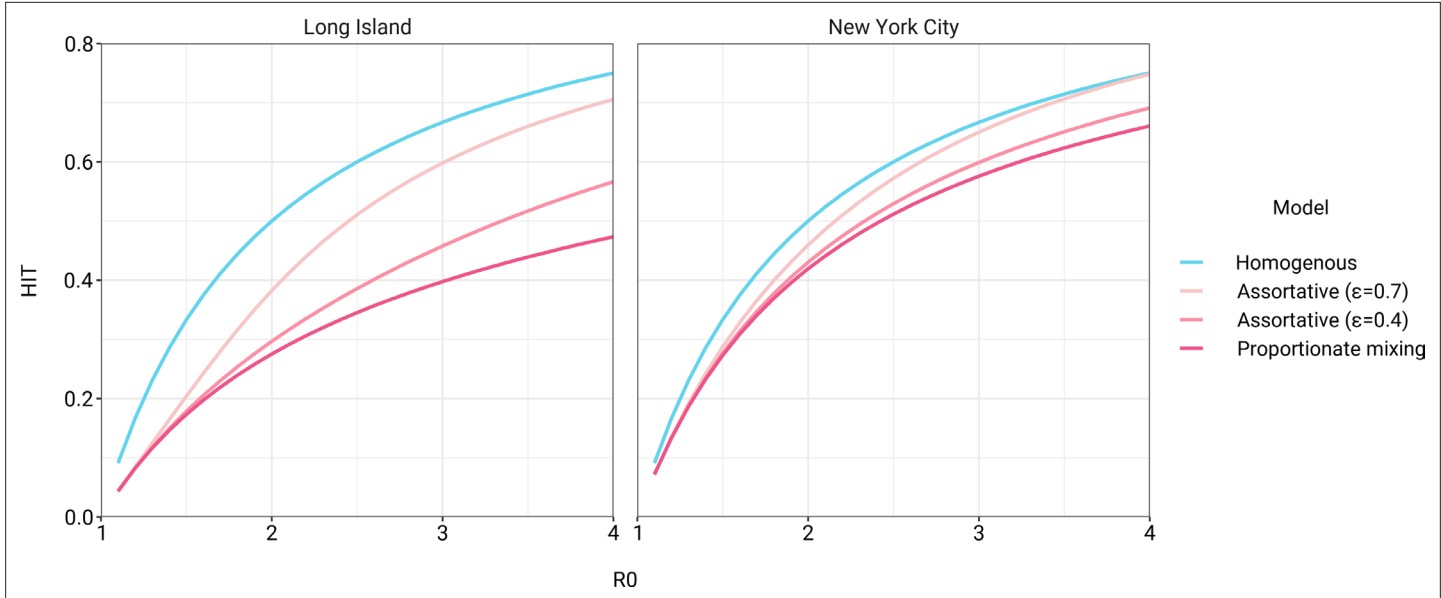

**Figure 1.** Incorporating assortativity in variable exposure models results in increased herd immunity thresholds across a range of $R_0$ values. Variable exposure models were fitted to New York City and Long Island serosurvey data.

The online version of this article includes the following figure supplement(s) for figure 1:

**Figure supplement 1.** $R_t$ calculation accounting for $\alpha$ (top) and without accounting for $\alpha$ (bottom) for a mitigated epidemic trajectory.

**Figure supplement 2.** Models incorporating variable susceptibility to COVID-19 fitted to New York City and Long Island serosurvey data result in reduced herd immunity thresholds (top) and final epidemic sizes (bottom) across a range of $R_0$ values.

**Figure supplement 3.** Comparison of herd immunity thresholds (HITs) from simulations to theoretical HIT curves for models with gamma distributed exposure and susceptibility (***Tkachenko et al., 2020***).

**Figure supplement 4.** Incorporating assortativity in variable exposure models results in increased final epidemic sizes across a range of $R_0$ values.

**Figure supplement 5.** Herd immunity thresholds versus $R_0$ in variable exposure models with mitigation measures for $\alpha = 0.3$ (top) and $\alpha = 0.6$ (bottom).

**Figure supplement 6.** Final epidemic sizes versus $R_0$ in variable exposure models with mitigation measures for $\alpha = 0.3$ (top) and $\alpha = 0.6$ (bottom).

**Figure supplement 7.** Sensitivity analysis on the impact of intensity and duration of non-pharmaceutical interventions (NPIs) on final epidemic sizes.

NYC. Model fits to Long Island resulted in even more pronounced exposure differences because of greater between-group differences in seropositivity (e.g., the seropositivity in Hispanics or Latinos relative to non-Hispanic whites was 1.85 times higher in Long Island than in NYC). Under proportionate mixing, Hispanics or Latinos, non-Hispanic Black people, non-Hispanic Asians, and multiracial or other people had 4.31, 1.96, 0.92, and 2.48 times the fitted total contact rates relative to non-Hispanic whites in Long Island, respectively. These differences in exposure impacted herd immunity levels and final epidemic sizes relative to the homogeneous model across a range of $R_0$ values (***Figure 1*** and ***Figure 1—figure supplement 4***); for example, for an $R_0$ of 3, the HIT decreases to 58% in NYC and 40% in Long Island compared to 67% under the homogeneous model. The observed contrast in HITs and final sizes between the proportionate mixing and the homogenous model is in line with theoretical derivations (***Figure 1—figure supplement 3***). The HIT overall is reached in this model after cumulative incidence has disproportionately increased in certain minority groups: at the HIT, 75% of Hispanics or Latinos and 63% of non-Hispanic Black people were infected compared to 46% of non-Hispanic whites in NYC, and 77% of Hispanics or Latinos and 48% of non-Hispanic Black people were infected compared to 29% of non-Hispanic whites in Long Island (***Figure 2***).

The estimated total contact rate ratios indicate increased contacts for minority groups such as Hispanics or Latinos and non-Hispanic Black people, which is in line with studies using cell phone mobility data (***Chang et al., 2021***); however, the magnitudes of the ratios are substantially higher than one would expect given the findings from those studies. This may reflect some of the limitations of the proportionate mixing assumption, which does not allow for preferential within-group contacts and hence must fit observed seropositivity differences solely by scaling total contact rates. To address this,

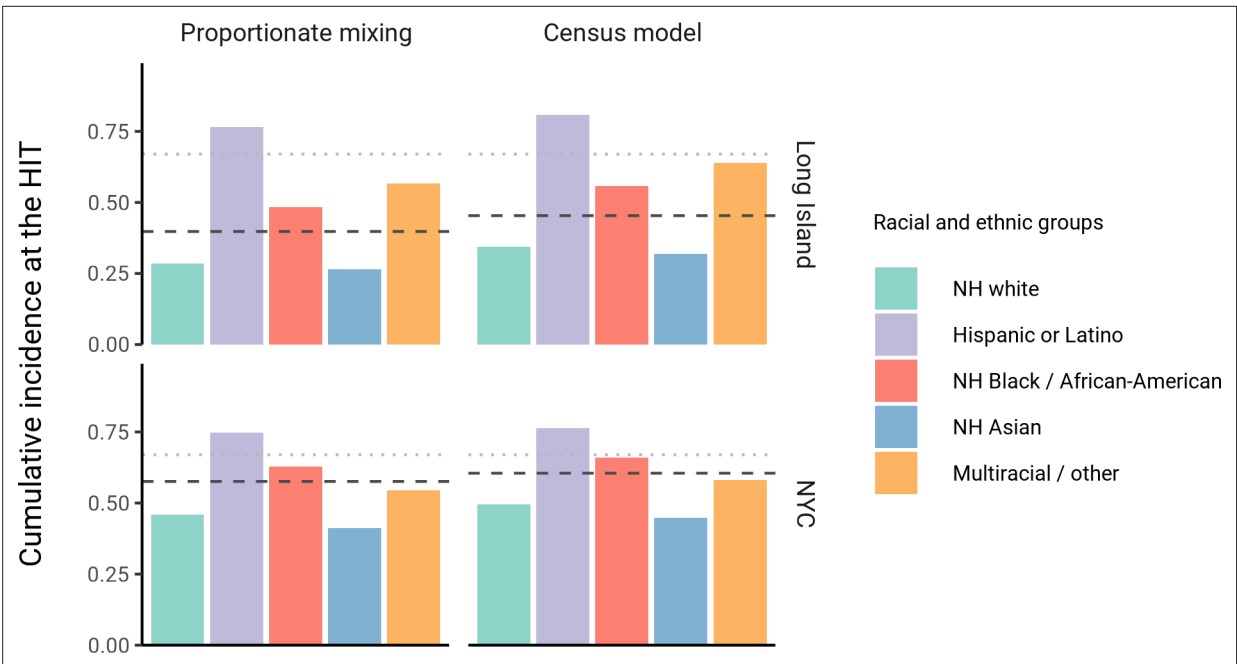

**Figure 2.** Cumulative incidence is disproportionately higher in some racial and ethnic minorities when the overall herd immunity threshold (HIT) is reached across model types and locations. Results are shown for an epidemic with $R_0$ = 3. The HIT for the population is indicated with a black line, and the HIT for a homogeneous model with the same $R_0$ is indicated with a gray line.

The online version of this article includes the following figure supplement(s) for figure 2:

**Figure supplement 1.** Sensitivity analysis on timing of the serosurvey relative to the start of the epidemic.

**Figure supplement 2.** Sensitivity analysis on initial number of infected individuals in each group.

**Figure supplement 3.** Sensitivity analysis on race or ethnicity of first infected individual.

**Figure supplement 4.** First iteration of fitting $\epsilon$ in social contact matrices to census data for New York City (top) and Long Island (bottom).

**Figure supplement 5.** Next-generation matrices for variable susceptibility (top), proportionate mixing (middle), and census-informed assortativity (bottom) models fitted to New York City seroprevalence data.

**Figure supplement 6.** Next-generation matrices for variable susceptibility (top), proportionate mixing (middle), and census-informed assortativity (bottom) models fitted to Long Island seroprevalence data.

**Figure supplement 7.** Distribution of census block group sizes in New York City (NYC; left) and Long Island (right).

**Figure supplement 8.** Cumulative incidence is disproportionately higher in some racial and ethnic minorities when the overall herd immunity threshold (HIT) is reached across model types and locations.

we augment the model by partitioning a specified fraction $\epsilon$ of contacts to be exclusively within-group, with the remaining contacts distributed proportionately. This assortative mixing model captures more realistic patterns of interactions due to neighborhood structure. After fitting the models across a range of $\epsilon$ values, we observed that as $\epsilon$ increases, HITs and epidemic final sizes shifted higher back towards the homogeneous case (*Figure 1*, *Figure 1—figure supplement 4*); this effect was less pronounced for $R_0$ values close to 1. This observation can be understood by comparing the epidemic cumulative incidence trajectories (*Figure 3—figure supplement 1*) and next-generation matrices (*Figure 2—figure supplement 5* and *6*): under proportionate mixing ($\epsilon = 0$), lower-risk demographic groups are protected from further infection due to built-up immunity in higher-risk demographic groups, but the magnitude of this indirect protection decreases as the proportion of exclusively within-group contacts increases and groups become more isolated.

We assessed a range of values for $\epsilon$ because the serosurvey data cannot be used to also fit the optimal $\epsilon$ value; given limited numbers of data points, any value of $\epsilon$ can fit exactly to the single seroprevalence time point we consider. To inform plausible assortativity levels, we instead used additional data on demographic population distributions from the American Community Survey US census stratified at the census block group level, which represents a small geographic area and population (*Figure 2—figure supplement 7*). We first calculated the exposure index, which represents the

average neighborhood's demographic composition from the perspective of an individual from a given racial or ethnic group; the proportion of contacts within group were elevated, suggesting assortativity in the census data (*Supplementary file 1*). To directly fit $\epsilon$ with these data, we assumed proportionate mixing within census block groups and ran an iterative fitting approach to jointly fit $\epsilon$ and the $a_i$ total contact rates using both census and serosurvey data (see Materials and methods and *Supplementary file 3*). This approach accounts for contacts based on geographic proximity using strong assumptions on mixing patterns, but may not capture contacts in other settings, such as work, beyond one's immediate neighborhood of residence. The models jointly fitted to serosurvey and census data indicated that 46 and 39% of contacts were exclusively within-group in NYC and Long Island, respectively. The same groups had elevated total contact rates as under proportionate mixing, but the magnitudes of differences were now lower and more concordant with reported mobility differences (*Chang et al., 2021*): model estimates indicated Hispanics or Latinos, non-Hispanic Black people, non-Hispanic Asians, and multiracial or other people had 1.62, 1.35, 0.90, and 1.17 times the total contact rate relative to non-Hispanic whites in NYC, respectively, and 2.60, 1.63, 0.93, and 1.90 times the total contact rate relative to non-Hispanic whites in Long Island, respectively. For an epidemic with $R_0 = 3$, the HITs for NYC and Long Island using these census-informed assortativity models were 61 and 45%, respectively. Similar to previous models, at the HIT, 76% of Hispanics or Latinos and 66% of non-Hispanic Black people were infected compared to 50% of non-Hispanic whites in NYC, and 81% of Hispanics or Latinos and 56% of non-Hispanic Black people were infected compared to 34% of non-Hispanic whites in Long Island (*Figure 2*).

Using these census-informed assortative mixing models, we then considered how the relative incidence rates of infection in demographic groups could change over the course of the epidemic. Early comparisons of infection and mortality rates have helped to identify racial and ethnic groups at high risk and the risk factors for infection (*Chamie et al., 2020*; *Moore et al., 2020*; *Millett et al., 2020a*; *Pan et al., 2020*; *Chen and Krieger, 2020*; *Bassett et al., 2020*; *Hanage et al., 2020*), but these studies often rely on cross-sectional snapshots of epidemiological patterns. The challenge is that these metrics can change over time: for instance, multiple studies indicate decreasing disparities in incidence rate over time across racial and ethnic groups relative to non-Hispanic whites (*Krieger*

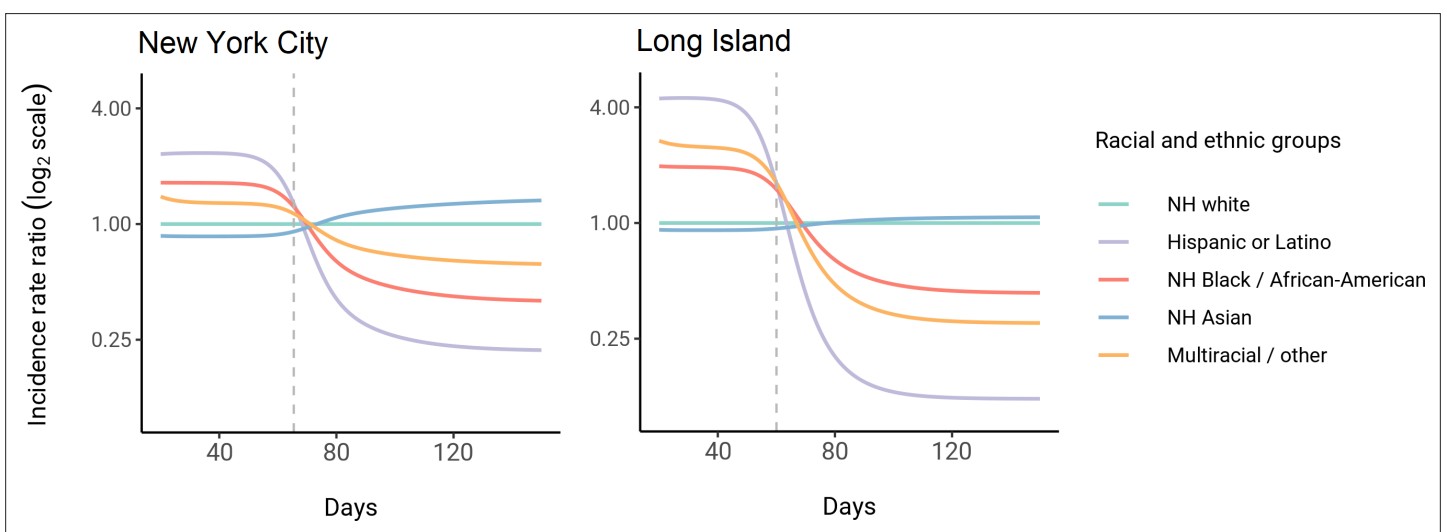

**Figure 3.** Dynamics of incidence rate ratios relative to non-Hispanic whites in assortative mixing models fitted to census and serosurvey data. Dashed line represents the peak overall incidence for the epidemic.

The online version of this article includes the following figure supplement(s) for figure 3:

**Figure supplement 1.** Comparison of cumulative incidence trajectories for proportionate mixing (top) and assortative mixing ($\epsilon = 0.7$; bottom) models fitted to Long Island seroprevalence data.

**Figure supplement 2.** Dynamics of cumulative incidence rate ratios relative to non-Hispanic whites in census-informed assortative mixing models, fitted to New York City (top) and Long Island (bottom) seroprevalence data.

**Figure supplement 3.** Dynamics of incidence rate ratios relative to non-Hispanic whites in census-informed assortative mixing models with mitigation measures.

*et al., 2020a*; *Van Dyke et al., 2021*). The reasons for these changes are multifaceted, but even independent of the effect of interventions, behavioral changes, or differential access to testing, models of epidemic spread in structured populations imply that incidence rate ratios for high-risk groups can decrease substantially as the epidemic progresses because of depletion of susceptible individuals from these groups (*Goldstein et al., 2017*; *Koopman et al., 1991*). In line with this, we observe that instantaneous incidence rate ratios are elevated initially in high-contact groups relative to non-Hispanic whites, but this trend reverses after the epidemic has peaked and overall incidence is decreasing – a consequence of the fact that a majority of individuals have already become infected (*Figure 3*). Similarly, cumulative incidence ratios remain elevated in high-contact racial and ethnic groups throughout the epidemic, but the magnitude decreases as the epidemic progresses (*Figure 3—figure supplement 2*). Although these trajectories are not meant to be taken as predictive estimates, these results highlight the importance of controlling for dynamics-induced changes in epidemiological measures of disease burden when evaluating the impact of interventions for reducing inequities in SARS-CoV-2 infections (*Kahn et al., 2020*); otherwise, ineffective interventions – depending on their timing – might still be associated with declines in relative measures of disease burden.

Finally, we assessed how robust these findings were to the impact of social distancing and other NPIs. We modeled these mitigation measures by scaling the transmission rate by a factor $\alpha$ beginning when 5% cumulative incidence in the population was reached. Setting the duration of distancing to be 50 days and allowing $\alpha$ to be either 0.3 or 0.6 (i.e., a 70% or 40% reduction in transmission rates, respectively), we assessed how the $R_0$ versus HIT and final epidemic size relationships changed. We found that the $R_0$ versus HIT relationships were similar comparing a mitigated to an unmitigated epidemic (*Figure 1—figure supplement 5*). In contrast, final epidemic sizes depended on the intensity of mitigation measures, though qualitative trends across models (e.g., increased assortativity leads to greater final sizes) remained true (*Figure 1—figure supplement 6*). To explore this further, we systematically varied $\alpha$ and the duration of NPIs while holding $R_0$ constant at 3. We found again that the HIT was consistent, whereas final epidemic sizes were substantially affected by the choice of mitigation parameters (*Figure 1—figure supplement 7*); the distribution of cumulative incidence at the point of HIT was also comparable with and without mitigation measures (*Figure 2—figure supplement 8*). The most stringent NPI intensities did not necessarily lead to the smallest epidemic final sizes, an idea which has been explored in studies analyzing optimal control measures (*Neuwirth et al., 2020*; *Handel et al., 2007*). Longitudinal changes in incidence rate ratios also were affected by NPIs, but qualitative trends in the ordering of racial and ethnic groups over time remained consistent (*Figure 3—figure supplement 3*).

## Discussion

Here, we explored how incorporating heterogeneity in SARS-CoV-2 spread across racial and ethnic groups could affect epidemic dynamics using deterministic transmission models. Models incorporating variable exposure generally decreased the HIT and final epidemic size, but incorporating preferential within-group contacts shifted HITs and final epidemic sizes higher, approaching the homogeneous case. Epidemiological measures of disease burden such as incidence rate ratios and cumulative incidence ratios also changed substantially over the course of the epidemic, highlighting the need to account for these trends when evaluating interventions (*Kahn et al., 2020*). These results illustrate the varied effects of different structured heterogeneity models, but are not meant to be best estimates given the limited seroprevalence data.

Across all model variants, the observed higher cumulative incidence among Hispanics or Latinos and non-Hispanic Black people compared to non-Hispanic whites led to estimates of higher estimated contacts relative to non-Hispanic whites, mirroring existing inequities in housing, education, healthcare, and beyond (*Khazanchi et al., 2020*; *Millett et al., 2020a*; *Millett et al., 2020b*; *Boyd et al., 2020*; *Bailey et al., 2021*). The estimated contact rate differences also concord with reports that frontline workers, who are unable to engage in physical or social distancing to the same degree as other types of workers, are disproportionately from minority backgrounds (*Blau et al., 2020*; *Chang et al., 2021*; *Kissler et al., 2020*). The assortativity we observed in the census data has root causes in many areas, including residential segregation arising from a long history of discriminatory practices (*Yang et al., 2020*; *Millett et al., 2020b*; *Bailey et al., 2021*; *Benfer et al., 2021*). Projecting the epidemic forward indicated that the overall HIT was reached after cumulative incidence had

increased disproportionately in minority groups, highlighting the fundamentally inequitable outcome of achieving herd immunity through infection. All of these factors underscore the fact that incorporating heterogeneity in models in a mechanism-free manner can conceal the disparities that underlie changes in epidemic final sizes and HITs. In particular, overall lower HIT and final sizes occur because certain groups suffer not only more infection than average, but more infection than under a homogeneous mixing model; incorporating heterogeneity lowers the HIT but increases it for the highest-risk groups (*Figure 2*).

These results also suggest that public health interventions for reducing COVID-19 inequities can have synergistic effects for controlling the overall epidemic (*Richardson et al., 2020*). For instance, from a transmission-control perspective, age-structured models indicate that vaccination of high-contact age groups – such as young adults – is optimal for controlling the spread of SARS-CoV-2 (*Bubar et al., 2021*). Similar interventions are being explored for disadvantaged populations because of how increased exposure underlies much of the higher infection risk that racial and ethnic minorities experience. For instance, Mulberry et al. used a structured SEIR model to evaluate the impact of preferentially vaccinating essential workers, finding that such a strategy was more effective in reducing morbidity, mortality, and economic cost due to COVID-19 compared to age-only vaccination prioritization strategies (*Mulberry et al., 2021*). Because racial and ethnic minorities are overrepresented among essential workers (*Blau et al., 2020*; *Chang et al., 2021*; *Kissler et al., 2020*), further modeling studies could evaluate the impact of such strategies on also reducing inequities in COVID-19 infection rates. Wrigley-Field et al. took a different methodological approach, projecting demographically stratified death rates from 2020 into 2021 assuming various vaccination strategies (*Wrigley-Field et al., 2021*). In line with the prior study, they found that vaccination strategies that prioritized geographic areas based on socioeconomic criteria or prior COVID mortality rates outperformed age-based strategies in reducing overall mortality and inequities in mortality. Because populations here were considered independently, these results could be extended using a transmission modeling framework, such as the one we have described, that accounts for interactions between racial and ethnic groups and thus allows for synergistic benefits from vaccinating high-risk populations. All policy proposals in this space should, of course, carefully consider the legal and ethical dimensions of vaccinations or other interventions targeted by race or ethnicity (*Schmidt et al., 2020*).

We note several limitations with this study. First, biases in the serosurvey sampling process can substantially affect downstream results; any conclusions drawn depend heavily on the degree to which serosurvey design and post-survey adjustments yield representative samples (*Clapham et al., 2020*). For instance, because the serosurvey we relied on primarily sampled people at grocery stores, there is both survival bias (cumulative incidence estimates do not account for people who have died) and ascertainment bias (undersampling of at-risk populations that are more likely to self-isolate, such as the elderly) (*Rosenberg et al., 2020*; *Accorsi et al., 2021*). These biases could affect model estimates if, for instance, the capacity to self-isolate varies by race or ethnicity – as suggested by associations of neighborhood-level mobility versus demographics (*Kishore et al., 2020a*; *Kissler et al., 2020*) – leading to an overestimate of cumulative incidence and contact rates in whites. Other sources of uncertainty, such as antibody test sensitivity and specificity, could also be incorporated into transmission models in future work (*Larremore et al., 2020*; *Accorsi et al., 2021*). Second, we have assumed that seropositivity implies complete immunity and that immunity does not wane. These are strong assumptions that can be revisited as empirical studies on the length of natural immunity are conducted. Third, we have assumed the impact of NPIs such as stay-at-home policies, closures, and the like to equally affect racial and ethnic groups. Empirical evidence suggests that during periods of lockdown certain neighborhoods that are disproportionately wealthy and white tend to show greater declines in mobility than others (*Kishore et al., 2020a*; *Kissler et al., 2020*). These simplifying assumptions were made to aid in illustrating the key findings of this model, but for more detailed predictive models, the extent to which contact rate differences change could be evaluated using longitudinal contact survey data (*Feehan and Mahmud, 2020*) since granular mobility data are typically not stratified by race and ethnicity due to privacy concerns (*Kishore et al., 2020b*). Fourth, due to data availability, we have only considered variability in exposure due to one demographic characteristic; models should ideally strive to also account for the effects of age on susceptibility and exposure within strata of race and ethnicity and other relevant demographics, such as socioeconomic status and occupation (*Mulberry et al., 2021*). These models could be fit using representative serological studies with detailed

cross-tabulated seropositivity estimates. Finally, we have estimated model parameters using a single cross-sectional serosurvey. To improve estimates and the ability to distinguish between model structures, future studies should use longitudinal serosurveys or case data stratified by race and ethnicity and corrected for underreporting; the challenge will be ensuring that such data are systematically collected and made publicly available, which has been a persistent barrier to research efforts (*Krieger et al., 2020b*). Addressing these data barriers will also be key for translating these and similar models into actionable policy proposals on vaccine distribution and NPIs.

In summary, we have explored how deterministic transmission models can be extended to study the dynamics of infection in racial and ethnic groups, and how the impact of heterogeneity on the HIT and final epidemic size depends strongly on the details of how heterogeneity is modeled. We have shown that due to early infections in individuals from the most at-risk group, relative measures of incidence may decline and even reverse, but inequities in the cumulative burden of infection persist throughout the epidemic as the HIT is reached. These results describe a framework that can be extended to other cities and countries in which racial and ethnic disparities in seropositivity have been observed (*Hallal et al., 2020*; *Flannery et al., 2020*; *Chan et al., 2021*) and are a step towards using transmission models to design policy interventions for reducing disparities in COVID-19 and other diseases.

## Acknowledgements

We thank Dr. Eli Rosenberg at the University at Albany School of Public Health and members of the Center for Communicable Disease Dynamics and Dr. Mary Bassett at the Harvard T.H. Chan School of Public Health for helpful comments. KCM was supported by the National Science Foundation GRFP grant DGE1745303. YHG and ML were funded by the Morris-Singer Foundation. M.L. was supported by SeroNet cooperative agreement U01 CA261277. Research reported in this publication was supported by the National Cancer Institute of the National Institutes of Health under Award Number U01CA261277. The content is solely the responsibility of the authors and does not necessarily represent the official views of the National Institutes of Health.

## Additional information

### Competing interests

Marc Lipsitch: Reviewing editor, *eLife*. The other authors declare that no competing interests exist.

### Funding

| Funder | Grant reference number | Author |
| --- | --- | --- |
| National Science Foundation | DGE1745303 | Kevin C Ma |
| Morris-Singer Foundation | | Yonatan H Grad Marc Lipsitch |
| National Cancer Institute | U01 CA261277 | Marc Lipsitch |

The funders had no role in study design, data collection and interpretation, or the decision to submit the work for publication.

### Author contributions

Kevin C Ma, Formal analysis, Investigation, Software, Visualization, Writing – original draft, Writing – review and editing; Tigist F Menkir, Investigation, Software, Validation, Writing – review and editing; Stephen Kissler, Formal analysis, Methodology, Writing – review and editing; Yonatan H Grad, Conceptualization, Funding acquisition, Resources, Supervision, Writing – review and editing; Marc Lipsitch, Conceptualization, Funding acquisition, Methodology, Supervision, Writing – review and editing

### Author ORCIDs

Kevin C Ma ![ORCID] http://orcid.org/0000-0002-4326-2911
Tigist F Menkir ![ORCID] http://orcid.org/0000-0001-6070-8017
Stephen Kissler ![ORCID] http://orcid.org/0000-0001-6000-8387

Yonatan H Grad  http://orcid.org/0000-0001-5646-1314
Marc Lipsitch  http://orcid.org/0000-0003-1504-9213

**Decision letter and Author response**
Decision letter https://doi.org/10.7554/eLife.66601.sa1
Author response https://doi.org/10.7554/eLife.66601.sa2

---

## Additional files

### Supplementary files
• Supplementary file 1. Exposure index matrices for New York City and Long Island.

• Supplementary file 2. Total contact rate ratios relative to non-Hispanic whites for proportionate mixing and census models fit to New York City and Long Island data.

• Supplementary file 3. Iterative census model fitting results for New York City and Long Island.

• Transparent reporting form

### Data availability
All data generated or analysed during this study are included in the manuscript and supporting files.

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
