## [Decision Letter]

**Acceptance summary:**

Thank you for submitting and revising this paper which makes the important point that the herd immunity threshold, as well as the pattern of incidence over time, is likely to be dictated by the degree of assortative mixing between racial and ethnic groups during the SARS-CoV-2 pandemic. The paper's observations can hopefully be used to optimize non-pharmaceutical interventions and vaccine implementation in high-risk ethnic and racial groups.

**Decision letter after peer review:**

Thank you for submitting your article "Modeling the impact of racial and ethnic disparities on COVID-19 epidemic dynamics" for consideration by *eLife*. Your article has been reviewed by 3 peer reviewers, including Joshua T Schiffer as the Reviewing Editor and Reviewer #1, by Miles Davenport as the Senior Editor. The following individuals involved in review of your submission have agreed to reveal their identity: Jon Zelner (Reviewer #2); Mia Moore (Reviewer #3).

The reviewers have discussed their reviews with one another in detail, and the Reviewing Editor has drafted this to help you prepare a revised submission.

Essential revisions:

1) Please consider broadening the range of R0 considered in the analysis, with inclusion of values closer to one. It would also be useful to allow R0 (or rather R_effective) to take on varying levels over time to reflect observed shifts in physical distancing and NPIs during the pandemic. All 3 reviewers agreed that it would be interesting to see whether these changes, which would more closely approximate reality, would alter the paper's conclusions in anyway.

We do not feel that it is required to precisely recapitulate the dynamics of the NYC and/or Long Island epidemics, but rather to attempt scenarios other than the unmitigated simulated epidemic in the original version.

2) Similarly, another way to strengthen the paper would be to show what happens when "herd immunity" is achieved via vaccination rather than infection. The interaction with the mixing models would be a fascinating and useful contribution.

*Reviewer #1 (Recommendations for the authors):*

Strengths:

1) The model structure is appropriate for the scientific question.

2) The paper addresses a critical feature of SARS-CoV-2 epidemiology which is its much higher prevalence in Hispanic or Latino and Black populations. In this sense, the paper has the potential to serve as a tool to enhance social justice.

3) Generally speaking, the analysis supports the conclusions.

Other considerations:

1) The clean distinction between susceptibility and exposure models described in the paper is conceptually useful but is unlikely to capture reality. Rather, susceptibility to infection is likely to vary more by age whereas exposure is more likely to vary by ethnic group / race. While age cohort are not explicitly distinguished in the model, the authors would do well to at least vary susceptibility across ethnic groups according to different age cohort structure within these groups. This would allow a more precise estimate of the true effect of variability in exposures. Alternatively, this could be mentioned as a limitation of the current model.

2) I appreciated that the authors maintained an agnostic stance on the actual value of HIT (across the population and within ethnic groups) based on the results of their model. If there was available data, then it might be possible to arrive at a slightly more precise estimate by fitting the model to serial incidence data (particularly sorted by ethnic group) over time in NYC and Long Island. First, this would give some sense of R_effective. Second, if successive waves were modeled, then the shift in relative incidence and CI among these groups that is predicted in Figure 3 and Sup Figure 8 may be observed in the actual data (this fits anecdotally with what I have seen in several states). Third, it may (or may not) be possible to estimate values of critical model parameters such as epsilon. It would be helpful to mention this as possible future work with the model.

Caveats about the impossibility of truly measuring HIT would still apply (due to new variants, shifting use and effective of NPIs, etc….). However, as is, the estimates of possible values for HIT are so wide as to make the underlying data used to train the model almost irrelevant. This makes the potential to leverage the model for policy decisions more limited.

3) I think the range of R0 in the figures should be extended to go as low as 1. Much of the pandemic in the US has been defined by local Re that varies between 0.8 and 1.2 (likely based on shifts in the degree of social distancing). I therefore think lower HIT thresholds should be considered and it would be nice to know how the extent of assortative mixing effects estimates at these lower R_e values.

4) line 274: I feel like this point needs to be considered in much more detail, either with a thoughtful discussion or with even with some simple additions to the model. How should these results make policy makers consider race and ethnicity when thinking about the key issues in the field right now such as vaccine allocation, masking, and new variants. I think to achieve the maximal impact, the authors should be very specific about how model results could impact policy making, and how we might lower the tragic discrepancies associated with COVID. If the model / data is insufficient for this purpose at this stage, then what type of data could be gathered that would allow more precise and targeted policy interventions?

*Reviewer #2 (Recommendations for the authors):*

Overall I think this is a solid and interesting piece that is an important contribution to the literature on COVID-19 disparities, even if it does have some limitations. To this point, most models of SARS-CoV-2 have not included the impact of residential and occupational segregation on differential group-specific covid outcomes. So, the authors are to commended on their rigorous and useful contribution on this valuable topic. I have a few specific questions and concerns, outlined below:

1. Does the reliance on serosurvey data collected in public places imply a potential issue with left-censoring, i.e. by not capturing individuals who had died? Can the authors address how survival bias might impact their results? I imagine this could bring the seroprevalence among older people down in a way that could bias their transmission rate estimates.

2. It might be helpful to think in terms of disparities in HITs as well as disparities in contact rates, since the HIT of whites is necessarily dependent on that of Blacks. I'm not really disagreeing with the thrust of what their analysis suggests or even the factual interpretation of it. But I do think it is important to phrase some of the conclusions of the model in ways that are more directly relevant to health equity, i.e. how much infection/vaccination coverage does each group need for members of that group to benefit from indirect protection?

3. The authors rely on a modified interaction index parameterized directly from their data. It would be helpful if they could explain why they did not rely on any sources of mobility data. Are these just not broken down along the type of race/ethnicity categories that would be necessary to complete this analysis? Integrating some sort of external information on mobility would definitely strengthen the analysis.

*Reviewer #3 (Recommendations for the authors):*

The authors show that in an uncontrolled epidemic the herd-immunity threshold may differ dramatically between racial groups. Although this question is undeniably important and the authors have shown that their results are robust to differing assumptions about population mixing, it seems to me that the situation considered is not completely relevant to current state of the covid epidemic. The herd-immunity threshold is assumed to be reached via a single, unmitigated wave within a few months. Unfortunately, the details of how this threshold is calculated are missing from the manuscript, but I can't help but imagine that it would be quite different if it was assumed to be possible to reach herd-immunity via vaccination. I think the authors need to address this.

Figure 2 could potentially include the cumulative incidence associated with R0=3 in a homogeneous model.

Multi-panel figures are unlabeled and split over more than one page, separated from their captions.

Issue with assortative mixing assumption: When you assume a = 1, you assume that everyone makes the same number of contacts. How, then, is one group defined to be at higher risk than another. Is a really 1 in this case?

Issue with assortative mixing derivation: I have a minor concern about the derivation, specifically how the contact matrix is fit. When I try to follow your steps and fill-in the gaps I get a slightly different result. So please either clarify your reasoning or fix the result.

Total i seen by a single j in one day = c_ij*Ni

Total i seen by all j in one day = c_ij*Ni*Nj=total j seen by all i in one day

Using the P_ij formulation (and please drop the P_ab notation, it is not consistently used and P_ij makes it easier to follow)

Total j seen by a single i in one day = P_ij % of people seen by group i = P_ij*ai=P_ij (as a=1)

Total j seen by all i in one day = P_ij*Ni

Therefore you want to minimize |c_ij*Ni*Nj-P_ij*Ni|=|(1-e)N_j*Ni/sum(N_k)+e*Ni*d_ij – Ni*Pij|

= |Ni((1-e)Nj/sum(N_k) + e*d_ij – P_ij)|

Which is only different from what you have in that Ni and Nj are switched in one term.

With a not equal to 1 (which you appear to use?) should be something like:

= |Ni((1-e)a_i a_j Nj/sum(N_k) + e*d_ij – a_i a_j sum_k(N_ik N_jk/sum(a_i N_lk)))|

Where the P_ij term is now weighted by a.

Calculation of HIT not described: I also feel like I'm missing something about how HIT is derived from the simulations. It sounds as though you used a computation rather than analytic approach, but I cannot find where it is spelled out. That seems important.

Fit to sero-prevalence data not shown or described: You initially list fitted activity levels, can you provide error bounds for these values? What is the nature of the data that is being fit? Is it only a single proportion collected at a single time? Can you show model fits? Unclear what was done here?

You also say "differences in exposure…are in line with theoretical derivations". I assume that what you meant is that differences in herd immunity levels and final epidemic sizes are in line with theoretical derivations, but the sentence doesn't read that way.

Please put activity levels in a table: These values (and uncertainty ranges) should be in a table where it is easy to compare between the models.

Fit of assortative mixing model not shown or described: In the results you say you first choose epsilon, then you fit a afterwards. However the epsilon fit is performed assuming that a is zero. Can you explain the reasoning here? I'm concerned that this methodology underestimates mixing (because if contact is heterogeneous by group, then you are more likely to contact individuals from the higher risk group than is strictly predicted by geographic distribution).

Incidence rate ratio plot is misleading: The incidence does not follow this pattern in proportionate mixing, and incidence is dropping in all groups. Furthermore this plot shows a time series which is completely unrealistic and wildly different from data. Most importantly, it seems to be wholly irrelevant to the main point of the paper.

---

## [Author Response]

1) Please consider broadening the range of R0 considered in the analysis, with inclusion of values closer to one.

We agree this would be of interest and have extended the range of R0 values. Figure 1 has been updated accordingly; we also updated the text with new findings: “After fitting the models across a range of ε values, we observed that as ε increases, HITs and epidemic final sizes shifted higher back towards the homogeneous case (Figure 1 and Figure 1—figure supplement 4); this effect was less pronounced for R_0_ values close to 1.”

It would also be useful to allow R0 (or rather R_effective) to take on varying levels over time to reflect observed shifts in physical distancing and NPIs during the pandemic. All 3 reviewers agreed that it would be interesting to see whether these changes, which would more closely approximate reality, would alter the paper's conclusions in anyway.We do not feel that it is required to precisely recapitulate the dynamics of the NYC and/or Long Island epidemics, but rather to attempt scenarios other than the unmitigated simulated epidemic in the original version.

We have conducted additional analyses exploring the important suggestion by the reviewers that social distancing could affect these conclusions. The text and figures have been updated accordingly:

“Finally, we assessed how robust these findings were to the impact of social distancing and other non-pharmaceutical interventions (NPIs). We modeled these mitigation measures by scaling the transmission rate by a factor α beginning when 5% cumulative incidence in the population was reached. Setting the duration of distancing to be 50 days and allowing α to be either 0.3 or 0.6 (i.e. a 70% or 40% reduction in transmission rates, respectively), we assessed how the R_0_ versus HIT and final epidemic size relationships changed. We found that the R_0_versus HIT relationship was similar to in the unmitigated epidemic (Figure 1—figure supplement 5). In contrast, final epidemic sizes depended on the intensity of mitigation measures, though qualitative trends across models (e.g. increased assortativity leads to greater final sizes) remained true (Figure 1—figure supplement 6). To explore this further, we systematically varied α and the duration of NPIs while holding R_0_ constant at 3. We found again that the HIT was consistent, whereas final epidemic sizes were substantially affected by the choice of mitigation parameters (Figure 1—figure supplement 7); the distribution of cumulative incidence at the point of HIT was also comparable with and without mitigation measures (Figure 2—figure supplement 8). The most stringent NPI intensities did not necessarily lead to the smallest epidemic final sizes, an idea which has been explored in studies analyzing optimal control measures (Neuwirth et al., 2020; Handel et al., 2007). Longitudinal changes in incidence rate ratios also were affected by NPIs, but qualitative trends in the ordering of racial and ethnic groups over time remained consistent (Figure 3—figure supplement 3).

2) Similarly, another way to strengthen the paper would be to show what happens when "herd immunity" is achieved via vaccination rather than infection. The interaction with the mixing models would be a fascinating and useful contribution.

We appreciate the reviewers interest in how the models we have presented could be extended to incorporate vaccination. Ultimately, we felt it would be irresponsible to provide preliminary estimates that could be misinterpreted as policy suggestions using a model that was not originally designed for such a nuanced question, especially because explicitly race-based approaches are legally unclear and likely controversial (Schmidt et al., 2020). Other studies have instead addressed this question extensively using proxies for race and ethnicity -- such as geography and essential worker status -- and we have expanded the Discussion with references to this emerging and important body of work.

“Similar interventions are being explored for disadvantaged populations because of how increased exposure underlies much of the higher infection risk that racial and ethnic minorities experience. For instance, Mulberry et al. used a structured SEIR model to evaluate the impact of preferentially vaccinating essential workers, finding that such a strategy was more effective in reducing morbidity, mortality, and economic cost due to COVID-19 compared to age-only vaccination prioritization strategies (Mulberry et al., 2021). Because racial and ethnic minorities are overrepresented among essential workers (Blau et al., 2020; Chang et al., 2020; Kissler et al., 2020), further modeling studies could evaluate the impact of such strategies on also reducing inequities in COVID-19 infection rates. Wrigley-Field et al. took a different methodological approach, projecting demographically-stratified death rates from 2020 into 2021 assuming various vaccination strategies (Wrigley-Field et al., 2021). In line with the prior study, they found that vaccination strategies that prioritized geographic areas based on socioeconomic criteria or prior COVID mortality rates outperformed age-based strategies in reducing overall mortality and inequities in mortality. Because populations here were considered independently, these results could be extended using a transmission modeling framework, such as the one we have described, that accounts for interactions between racial and ethnic groups and thus allows for synergistic benefits from vaccinating high-risk populations. All policy proposals in this space should, of course, carefully consider the legal and ethical dimensions of vaccinations or other interventions targeted by race or ethnicity (Schmidt et al., 2020).”

Reviewer #1 (Recommendations for the authors):Strengths:1) The model structure is appropriate for the scientific question.2) The paper addresses a critical feature of SARS-CoV-2 epidemiology which is its much higher prevalence in Hispanic or Latino and Black populations. In this sense, the paper has the potential to serve as a tool to enhance social justice.3) Generally speaking, the analysis supports the conclusions.Other considerations:1) The clean distinction between susceptibility and exposure models described in the paper is conceptually useful but is unlikely to capture reality. Rather, susceptibility to infection is likely to vary more by age whereas exposure is more likely to vary by ethnic group / race. While age cohort are not explicitly distinguished in the model, the authors would do well to at least vary susceptibility across ethnic groups according to different age cohort structure within these groups. This would allow a more precise estimate of the true effect of variability in exposures. Alternatively, this could be mentioned as a limitation of the current model.

We agree that this would be an important extension for future work and have indicated this in the Discussion, along with the types of data necessary to fit such models:

“Fourth, due to data availability, we have only considered variability in exposure due to one demographic characteristic; models should ideally strive to also account for the effects of age on susceptibility and exposure within strata of race and ethnicity and other relevant demographics, such as socioeconomic status and occupation (Mulberry et al., 2021). These models could be fit using representative serological studies with detailed cross-tabulated seropositivity estimates.”

2) I appreciated that the authors maintained an agnostic stance on the actual value of HIT (across the population and within ethnic groups) based on the results of their model. If there was available data, then it might be possible to arrive at a slightly more precise estimate by fitting the model to serial incidence data (particularly sorted by ethnic group) over time in NYC and Long Island. First, this would give some sense of R_effective. Second, if successive waves were modeled, then the shift in relative incidence and CI among these groups that is predicted in Figure 3 and Sup Figure 8 may be observed in the actual data (this fits anecdotally with what I have seen in several states). Third, it may (or may not) be possible to estimate values of critical model parameters such as epsilon. It would be helpful to mention this as possible future work with the model.Caveats about the impossibility of truly measuring HIT would still apply (due to new variants, shifting use and effective of NPIs, etc….). However, as is, the estimates of possible values for HIT are so wide as to make the underlying data used to train the model almost irrelevant. This makes the potential to leverage the model for policy decisions more limited.

We have highlighted this important limitation in the Discussion:

“Finally, we have estimated model parameters using a single cross-sectional serosurvey. To improve estimates and the ability to distinguish between model structures, future studies should use longitudinal serosurveys or case data stratified by race and ethnicity and corrected for underreporting; the challenge will be ensuring that such data are systematically collected and made publicly available, which has been a persistent barrier to research efforts (Krieger et al., 2020). Addressing these data barriers will also be key for translating these and similar models into actionable policy proposals on vaccine distribution and non-pharmaceutical interventions.”

3) I think the range of R0 in the figures should be extended to go as as low as 1. Much of the pandemic in the US has been defined by local Re that varies between 0.8 and 1.2 (likely based on shifts in the degree of social distancing). I therefore think lower HIT thresholds should be considered and it would be nice to know how the extent of assortative mixing effects estimates at these lower R_e values.

Thanks for this suggestion; we agree -- see Essential Revisions point 1 for our response.

4) line 274: I feel like this point needs to be considered in much more detail, either with a thoughtful discussion or with even with some simple additions to the model. How should these results make policy makers consider race and ethnicity when thinking about the key issues in the field right now such as vaccine allocation, masking, and new variants. I think to achieve the maximal impact, the authors should be very specific about how model results could impact policy making, and how we might lower the tragic discrepancies associated with COVID. If the model / data is insufficient for this purpose at this stage, then what type of data could be gathered that would allow more precise and targeted policy interventions?

We agree with these points -- see Essential Revisions point 2 for our response, and points 1 and 2 above for examples of additional data required.

Reviewer #2 (Recommendations for the authors):Overall I think this is a solid and interesting piece that is an important contribution to the literature on COVID-19 disparities, even if it does have some limitations. To this point, most models of SARS-CoV-2 have not included the impact of residential and occupational segregation on differential group-specific covid outcomes. So, the authors are to commended on their rigorous and useful contribution on this valuable topic. I have a few specific questions and concerns, outlined below:

We thank the reviewer for the supportive comments.

1. Does the reliance on serosurvey data collected in public places imply a potential issue with left-censoring, i.e. by not capturing individuals who had died? Can the authors address how survival bias might impact their results? I imagine this could bring the seroprevalence among older people down in a way that could bias their transmission rate estimates.

We have included this important point in the limitations section on potential serosurvey biases: “First, biases in the serosurvey sampling process can substantially affect downstream results; any conclusions drawn depend heavily on the degree to which serosurvey design and post-survey adjustments yield representative samples (Clapham et al., 2020). For instance, because the serosurvey we relied on primarily sampled people at grocery stores, there is both survival bias (cumulative incidence estimates do not account for people who have died) and ascertainment bias (undersampling of at-risk populations that are more likely to self-isolate, such as the elderly; Rosenberg et al., 2020; Accorsi et al., 2021). These biases could affect model estimates if, for instance, the capacity to self-isolate varies by race or ethnicity -- as suggested by associations of neighborhood-level mobility versus demographics (Kishore et al., 2020; Kissler et al., 2020) -- leading to an overestimate of cumulative incidence and contact rates in whites.”

2. It might be helpful to think in terms of disparities in HITs as well as disparities in contact rates, since the HIT of whites is necessarily dependent on that of Blacks. I'm not really disagreeing with the thrust of what their analysis suggests or even the factual interpretation of it. But I do think it is important to phrase some of the conclusions of the model in ways that are more directly relevant to health equity, i.e. how much infection/vaccination coverage does each group need for members of that group to benefit from indirect protection?

We agree with this important point and indeed this was the goal, in part, of the analyses in Figure 2. We have added additional text to the Discussion highlighting this: “Projecting the epidemic forward indicated that the overall HIT was reached after cumulative incidence had increased disproportionately in minority groups, highlighting the fundamentally inequitable outcome of achieving herd immunity through infection. All of these factors underscore the fact that incorporating heterogeneity in models in a mechanism-free manner can conceal the disparities that underlie changes in epidemic final sizes and HITs. In particular, overall lower HIT and final sizes occur because certain groups suffer not only more infection than average, but more infection than under a homogeneous mixing model; incorporating heterogeneity lowers the HIT but increases it for the highest-risk groups (Figure 2).”

For vaccination, see our response to Essential Revisions point 2.

3. The authors rely on a modified interaction index parameterized directly from their data. It would be helpful if they could explain why they did not rely on any sources of mobility data. Are these just not broken down along the type of race/ethnicity categories that would be necessary to complete this analysis? Integrating some sort of external information on mobility would definitely strengthen the analysis.

This is a great suggestion, but this type of data has generally not been available due to privacy concerns from disaggregating mobility data by race and ethnicity (Kishore et al., 2020). Instead, we modeled NPIs as mentioned in Essential Revisions point 2, with the caveat that reduction in mobility was assumed to be identical across groups. We added this into the text explicitly as a limitation: “Third, we have assumed the impact of non-pharmaceutical interventions such as stay-at-home policies, closures, and the like to equally affect racial and ethnic groups. Empirical evidence suggests that during periods of lockdown, certain neighborhoods that are disproportionately wealthy and white tend to show greater declines in mobility than others (Kishore et al., 2020; Kissler et al., 2020). These simplifying assumptions were made to aid in illustrating the key findings of this model, but for more detailed predictive models, the extent to which activity level differences change could be evaluated using longitudinal contact survey data (Feehan and Mahmud, 2020), since granular mobility data are typically not stratified by race and ethnicity due to privacy concerns (Kishore et al., 2020).”

Reviewer #3 (Recommendations for the authors):The authors show that in an uncontrolled epidemic the herd-immunity threshold may differ dramatically between racial groups. Although this question is undeniably important and the authors have shown that their results are robust to differing assumptions about population mixing, it seems to me that the situation considered is not completely relevant to current state of the covid epidemic. The herd-immunity threshold is assumed to be reached via a single, unmitigated wave within a few months. Unfortunately, the details of how this threshold is calculated are missing from the manuscript, but I can't help but imagine that it would be quite different if it was assumed to be possible to reach herd-immunity via vaccination. I think the authors need to address this.

We have broadened the applicability of the findings by incorporating the effect of NPIs; see Essential Revisions points 1 and 2 for our responses.

Additionally, we have expanded the Methods section to explicitly state how we calculated the HIT and final epidemic size: “*Calculating the HIT and final epidemic size*”.

We defined the herd immunity threshold (HIT) for all models as the fraction of non-susceptible people when the effective reproduction number R_t_ first crosses 1. In the homogenous model, where R_t_ = S_t_ β / γ, the analytical solution for the HIT occurs when the fraction of non-susceptible individuals equals 1-1/R_0_. In the structured models of heterogeneous populations, the HIT was calculated via simulation: we took the dominant eigenvalue of G_t_ at each timestep to calculate R_t_, and identified the number of non-susceptible individuals when R_t_ first decreased below 1. For the heterogeneous models with mitigation measures, R_t_ was calculated at each timestep with respect to the corresponding unmitigated epidemic; in other words, the mitigation scaling factor α was not included in the R_t_ calculation. This identifies the point in the epidemic trajectory at which the population reaches the HIT even if all mitigation measures were lifted (i.e. HIT due to population immunity), as opposed to the point in the trajectory when the population transiently reaches the HIT due to mitigation measures (see Figure 1—figure supplement 1 for further explanation).

Final epidemic sizes were calculated by simulation by running the epidemic out to 1 year for R_0_ above 2, and 4 years for R_0_ below 2 to allow additional time for the slower epidemics to fully resolve. The final time point was used as the estimate for the final epidemic size.”

Figure 2 could potentially include the cumulative incidence associated with R0=3 in a homogeneous model.

We agree this could be helpful for readers and have updated Figure 2 and the caption accordingly.

Multi-panel figures are unlabeled and split over more than one page, separated from their captions.

Supplementary Figures 2 and 3 (now labelled as Figure 2—figure supplements 5 and 6) have been updated to be on one page with appropriate labels.

Issue with assortative mixing assumption: When you assume a = 1, you assume that everyone makes the same number of contacts. How, then, is one group defined to be at higher risk than another. Is a really 1 in this case?Issue with assortative mixing derivation: I have a minor concern about the derivation, specifically how the contact matrix is fit. When I try to follow your steps and fill-in the gaps I get a slightly different result. So please either clarify your reasoning or fix the result.Total i seen by a single j in one day = c_ij*NiTotal i seen by all j in one day = c_ij*Ni*Nj=total j seen by all i in one dayUsing the P_ij formulation (and please drop the P_ab notation, it is not consistently used and P_ij makes it easier to follow)Total j seen by a single i in one day = P_ij % of people seen by group i = P_ij*ai=P_ij (as a=1)Total j seen by all i in one day = P_ij*NiTherefore you want to minimize |c_ij*Ni*Nj-P_ij*Ni|=|(1-e)N_j*Ni/sum(N_k)+e*Ni*d_ij – Ni*Pij|= |Ni((1-e)Nj/sum(N_k) + e*d_ij – P_ij)|Which is only different from what you have in that Ni and Nj are switched in one term.With a not equal to 1 (which you appear to use?) should be something like:= |Ni((1-e)a_i a_j Nj/sum(N_k) + e*d_ij – a_i a_j sum_k(N_ik N_jk/sum(a_i N_lk)))|Where the P_ij term is now weighted by a.Fit of assortative mixing model not shown or described: In the results you say you first choose epsilon, then you fit a afterwards. However the epsilon fit is performed assuming that a is zero. Can you explain the reasoning here? I'm concerned that this methodology underestimates mixing (because if contact is heterogeneous by group, then you are more likely to contact individuals from the higher risk group than is strictly predicted by geographic distribution).

We thank the reviewer for working through the derivation -- we have improved our approach to fitting the census-informed model (please see Methods section “Model fitting and data sources” beginning with “We acquired total population numbers” for typeset equations) and updated all figures and analyses accordingly; the new model fitting approach should address the reviewer’s concerns because we now jointly fit epsilon and the a’s to the serosurvey and census data, and no longer are making assumptions about all a’s being 1.

Calculation of HIT not described: I also feel like I'm missing something about how HIT is derived from the simulations. It sounds as though you used a computation rather than analytic approach, but I cannot find where it is spelled out. That seems important.

HIT methodological details have now been provided (see above).

You also say "differences in exposure…are in line with theoretical derivations". I assume that what you meant is that differences in herd immunity levels and final epidemic sizes are in line with theoretical derivations, but the sentence doesn't read that way.

Thank you for noting this; the sentence has been updated: “The observed contrast in HITs and final sizes between the proportionate mixing and the homogenous model is in line with theoretical derivations (Figure 1—figure supplement 3).”

Fit to sero-prevalence data not shown or described: You initially list fitted activity levels, can you provide error bounds for these values? What is the nature of the data that is being fit? Is it only a single proportion collected at a single time? Can you show model fits? Unclear what was done here?

We have clarified that we are fitting to a single cross-sectional data point: “We estimated the a_i_ in the variable exposure models and the q_i_ in the variable susceptibility models using maximum likelihood fits to a single cross-sectional serosurvey from New York, which was collected from over 15,000 adults in grocery stores from April 19-28th (Rosenberg et al., 2020).”

The model fitting procedure is described in the “Model fitting and data sources” section of the Methods. Because of the nature of the data, there are no error bounds (all models tested fit exactly to the single time point), and this is now highlighted as an opportunity for improvement in future studies: “Finally, we have estimated model parameters using a single cross-sectional serosurvey. To improve estimates and the ability to distinguish between model structures, future studies should use longitudinal serosurveys or case data stratified by race and ethnicity and corrected for underreporting; the challenge will be ensuring that such data are systematically collected and made publicly available, which has been a persistent barrier to research efforts (Krieger et al., 2020).”

Please put activity levels in a table: These values (and uncertainty ranges) should be in a table where it is easy to compare between the models.

Activity levels (i.e. total contact rates) have been formatted into Supplementary File 2.

Incidence rate ratio plot is misleading: The incidence does not follow this pattern in proportionate mixing, and incidence is dropping in all groups.

The trend also holds true for proportionate mixing; for example, in Author response image 1 is the trajectory for the Long Island proportionate mixing model.

**Author response image 1. sa2fig1:** 

We have clarified that incidence is dropping in all groups: “In line with this, we observe that instantaneous incidence rate ratios are elevated initially in high-exposure contact groups relative to non-Hispanic whites, but this trend reverses after the epidemic has peaked and overall incidence is decreasing -- a consequence of the fact that a majority of individuals have already become infected.”

Furthermore this plot shows a time series which is completely unrealistic and wildly different from data. Most importantly, it seems to be wholly irrelevant to the main point of the paper.

We have clarified that our estimates are meant to be illustrative: “Although these trajectories are not meant to be taken as predictive estimates, these results highlight the importance of controlling for dynamics-induced changes in epidemiological measures of disease burden when evaluating the impact of interventions for reducing inequities in SARS-CoV-2 infections (Kahn et al., 2020)”

Taken qualitatively, we argue that these trends are in fact congruent with multiple reports of decreasing disparities in incidence rate over time across racial and ethnic groups. For instance, Van Dyke et al. showed that incidence in minority groups aged <25 years relative to non-Hispanic whites decreased substantially over the course of the year. These results are similar to county-level correlations reported by Krieger et al. that showed decreasing correlation over time of incidence with county percent of color.

The cause of these trends is an open question. Proposed explanations include differential access to testing, differences in adherence to distancing or mask wearing, and so on, but the possible contribution of intrinsic epidemic dynamics has not been addressed. By showing that our modeling framework can reproduce these trends, we have demonstrated another plausible explanation for the observed trends and highlighted the utility of using transmission models stratified by race and ethnicity. Additionally, these questions are not just theoretical: health equity metrics are being used to guide reopening policies, and it is important to understand how these epidemiological measures are expected to change over time as the epidemic progresses.